# Immunophenotypic but Not Genetic Changes Reclassify the Majority of Relapsed/Refractory Pediatric Cases of Early T-Cell Precursor Acute Lymphoblastic Leukemia

**DOI:** 10.3390/ijms25115610

**Published:** 2024-05-21

**Authors:** Irina Demina, Aya Dagestani, Aleksandra Borkovskaia, Alexandra Semchenkova, Olga Soldatkina, Svetlana Kashpor, Yulia Olshanskaya, Julia Roumiantseva, Alexander Karachunskiy, Galina Novichkova, Michael Maschan, Elena Zerkalenkova, Alexander Popov

**Affiliations:** Dmitry Rogachev National Medical Research Center of Pediatric Hematology, Oncology and Immunology, Moscow 117998, Russia; idemina@mail.ru (I.D.); ayadag@mail.ru (A.D.); aleksandra.borkovskaia@fccho-moscow.ru (A.B.); semalex94@mail.ru (A.S.); olga.soldatkina@fccho-moscow.ru (O.S.); kashpor_s@mail.ru (S.K.); yuliaolshanskaya@gmail.com (Y.O.); j.roumiantseva@mail.ru (J.R.); aikarat@mail.ru (A.K.); gnovichkova@yandex.ru (G.N.); mmaschan@yandex.ru (M.M.); eazerkalenkova@gmail.com (E.Z.)

**Keywords:** ETP-ALL, immunophenotype, relapse, lineage switch

## Abstract

Early T-cell precursor acute lymphoblastic leukemia (ETP-ALL) develops from very early cells with the potential for both T-cell and myeloid differentiation. The ambiguous nature of leukemic blasts in ETP-ALL may lead to immunophenotypic alterations at relapse. Here, we address immunophenotypic alterations and related classification issues, as well as genetic features of relapsed pediatric ETP-ALL. Between 2017 and 2022, 7518 patients were diagnosed with acute leukemia (AL). In addition to conventional immunophenotyping, karyotyping, and FISH studies, we performed next-generation sequencing of the T-cell receptor clonal repertoire and reverse transcription PCR and RNA sequencing for patients with ETP-ALL at both initial diagnosis and relapse. Among a total of 534 patients diagnosed with T-cell ALL (7.1%), 60 had ETP-ALL (11.2%). Ten patients with ETP-ALL experienced relapse or progression on therapy (16.7%), with a median time to event of 5 months (ranging from two weeks to 5 years). Most relapses were classified as AL of ambiguous lineage (n = 5) and acute myeloid leukemia (AML) (n = 4). Major genetic markers of leukemic cells remained unchanged at relapse. Of the patients with relapse, four had polyclonal leukemic populations and a relapse with AML or bilineal mixed-phenotype AL (MPAL). Three patients had clonal *TRD* rearrangements and relapse with AML, undifferentiated AL, or retention of the ETP-ALL phenotype. ETP-ALL relapse requires careful clinical and laboratory diagnosis. Treatment decisions should rely mainly on initial examination data, taking into account both immunophenotypic and molecular/genetic characteristics.

## 1. Introduction

Early T-cell precursor acute lymphoblastic leukemia (ETP-ALL) was initially introduced by E. Coustan-Smith et al. in 2009 [1] and established as a distinct type of T-cell ALL by the World Health Organization (WHO) in the 2016 revision [2] due to its unique immunophenotypic and genomic profiles. The gene signature of ETP-ALL differs from that of other T-cell ALL and resembles that of early T-lineage precursors [1,3]. In fact, ETP-ALL is a tumor that originates from immature hematopoietic progenitors and retains the capacity for both T-lineage and myeloid differentiation [4].

According to various estimates, the incidence of ETP-ALL in children is 5–17% [5,6,7,8]. This type of leukemia is usually characterized by genetic heterogeneity, poor response to chemotherapy, and a high risk of relapse [1,6,7,8,9].

Immunophenotyping by multicolor flow cytometry is the cornerstone method for ETP-ALL diagnosis [1,10]. Current immunophenotypic criteria for its confirmation include (1) verified T-lineage origin of leukemic cells (presence of intracellular CD3 and surface CD7); (2) absence of CD1a and CD8; (3) low expression of CD5 (<75% CD5-positive leukemic cells); and (4) presence of at least one of the antigens CD11b, CD13, CD33, CD34, CD117 and HLA-DR [10].

The dual character of ETP-ALL, which lacks definitive genetic features, creates the possibility of immunophenotypic alterations at relapse [11,12]. These alterations, however slight, can lead to reclassification of the disease. Thus, if intracellular CD3 is lost at relapse while no significant number of surface T-cell markers are detected at diagnosis, ETP-ALL should be reclassified to AML or acute leukemia of ambiguous lineage (ALAL).

Our work focuses on exploring issues of classification, immunophenotypic alterations, and the genetic landscape involved in relapse of ETP-ALL in pediatric patients.

## 2. Results

### 2.1. Group Description

Overall, 534 patients (7.1%) between 0 and 18 years of age were diagnosed with T-cell ALL. Of these, 60 patients (median age 9 years), including 42 boys and 18 girls (sex ratio 2.3:1), had ETP-ALL (11.2% of T-cell ALL cases, 0.8% of all cases). Ten patients (median age 7 years, range 6 months–16 years) with ETP-ALL experienced disease recurrence (1.9% of T-cell ALL cases, 16.7% of ETP-ALL cases), including eight boys and two girls (sex ratio 4:1).

### 2.2. Immunophenotyping 

Of 10 cases of ETP-ALL relapse/progression, only one case fully retained the original TII immunophenotype. In the remaining cases, the immunophenotype changed predominantly to AML (four cases) but also to bilineal MPAL (three cases), unclassifiable AL (one case), and undifferentiated AL (one case). The pattern of ETP-ALL relapse is shown in Figure 1 and Table 1.

Three of ten patients were initially diagnosed with TI-ALL, and seven had TII-ALL. According to the current diagnostic standards [13], TII-ALL was diagnosed based on the presence of surface CD2 and/or CD5 (CD5 expression level < 75%). Thus, the TII-ALL group of ETP-ALL can be divided into two subgroups based on the absence/presence of CD5. Four patients had CD5-negative leukemic blasts (with a median iCD3 expression of 31%), and three patients had CD5-positive blasts (with a median iCD3 expression of 58%) (Figure 1A).

Eight of ten patients experienced disease recurrence within the first year after diagnosis (including one patient with AML who had tumor progression in 25 days), one patient had a relapse with bilineal MPAL after 1 year, and one patient had a relapse with AML after 5 years (Figure 1B).

### 2.3. Genetic Studies 

Cytogenetic and molecular studies were carried out on the relapsed ETP cohort (n = 10) and a comparison cohort of ETP patients who had no evidence of relapse at the time of the study (n = 50). The median follow-up time was comparable between the two cohorts (5.0 y (1.5–11.3 y) vs. 3.6 y (1.2–11.3 y), *p* = 0.513).

The entire ETP-ALL group was genetically heterogeneous (Figure 2A). Recurrent genetic lesions were observed in 45% of cases (27 of 60) and included *BCL11B* locus (n = 10), *KMT2A* (n = 4), *ETV6* (n = 3), *RUNX1* (n = 3), *TLX3* locus (n = 3) and *HOXA* locus (n = 2) rearrangements. *TCRα*/*δ* and *TCRβ* locus rearrangements and *PICALM::MLLT10* fusion gene formation were also found (Appendix A). Ten patients exhibited a normal karyotype. Others carried nonspecific structural rearrangements and aneuploidies such as trisomy 8, trisomy 4, trisomy 10, and others (n = 20) or complex karyotypes (n = 3). Among the patients with relapse, single cases of t(7;12)(q36;p13)/*MNX1::ETV6* and t(9;11)(p21;q23)/*KMT2A::MLLT3* translocations and *BCL11B*, *TLX3* and *TCRα/δ* loci rearrangements were observed as well as complex karyotype (n = 2) and non-recurrent aberrations (n = 3). None had a normal karyotype at the initial examination. Thus, no translocations correlating with the disease recurrence were identified in our study.

The assessment of the *TCR*/*BCR* repertoire of the entire ETP-ALL group revealed that only half of the patients carried clonal rearrangements (24 of 42), while others exhibited polyclonal repertoire (18 of 42; Figure 2A). The majority of clonal *TCR* rearrangements found in patients were related to the *TRD* locus (18 of 24). However, the relapsed ETP cohort was not enriched in either of the patterns –3 of 7 patients exhibited *TCR* clonal rearrangements, which held true for 15 of 29 non-relapsed cohorts (*p* = 0.673, χ^2^-test). Thus, no *TCR*/*BCR* repertoire pattern correlating with disease recurrence was identified in our study.

The molecular features of leukemic cells at initial diagnosis and relapse of ETP-ALL are presented in Table 2. In most cases of relapse, the main genetic lesions remained stable throughout the course of the disease, either recurrent chromosomal aberrations (*KMT2A*, *ETV6*, *BCL11B*, *TLX3* rearrangements in cases 2, 4, 5, and 8, respectively) or aneuploidies (trisomy 8 and 10 in cases 3 and 7, respectively). Only *KMT2A::MLLT3*-positive ETP-ALL demonstrated clonal progression with an accumulation of additional chromosomal abnormalities (Appendix A). *TRD* clonal rearrangements in cases 1, 2, and 7 were also stable, and patients with an initial polyclonal pattern retained the polyclonal pattern (Appendix A). Regarding the correlation between clonal rearrangement patterns, we observed that ETP-ALL with a polyclonal *TCR*/*BCR* repertoire tended to relapse as AML (Figure 2B).

## 3. Discussion

ETP-ALL in children is considered unfavorable in terms of response to standard polychemotherapy and survival prognosis [1,6,8,9,14], though in modern trials, the outcome in these patients is not definitely poor [15,16]. This is mainly linked to the potential of leukemic cells for differentiation in both myeloid and T-lymphoid lineages. Such immaturity also results in similarities in the biological features of ETP-ALL and the T/Myelo type of MPAL [4,17,18]. In addition, the great lineage plasticity of early T-lineage leukemic blasts allows them to escape lymphoid-directed chemotherapy by the early switch to AML, which was demonstrated previously [11,12]. The relapse rate of ETP-ALL in our study was 16.7%.

As ETP-ALL by definition establishes not only in CD5-negative cases but also in cases with weak CD5 expression [1,10], this diagnosis can be made not only with the earliest TI EGIL subtype but also the more “mature” TII variant. The CD5-CD2+ subset of TII patients also belongs to this category. Indeed, the majority of ETP-ALL cases are formally diagnosed as TII (61.7% in our group). A similar distribution was observed in patients who experienced relapse/progression. In fact, the more “mature” subvariant of T-ALL in the ETP-ALL cohort did not affect lineage plasticity, and all except one studied patient experienced relapse/progression with a diagnosis other than T-ALL. It is necessary to determine that such a “lineage switch” is sometimes only formal, without real changes in leukemic cells. Surface markers of the T and myeloid lineages are often found simultaneously at diagnosis. This is because early T-cell precursors are essentially immature thymocytes that retain stem cell-like features and can differentiate into both T lymphoid and myeloid cells [19,20,21]. Their assignment to ETP-ALL by immunophenotyping is based on the detection of intracellular CD3 expression and the presence of surface CD7 [10,13]. According to the classification, 10% expression of intracellular CD3 is sufficient to assign the whole population to the T-cell lineage of differentiation [10]. However, in this case, 90% of leukemic cells lack this marker and, in fact, have myeloid features. If T-lineage markers fall below the threshold or disappear, such leukemia formally becomes myeloid, unclassifiable, or mixed-phenotypic per accepted classification. If leukemic cells have weak or negative CD5 at diagnosis and then lose intracellular CD3 at relapse, the leukemia should be reclassified as myeloid (Figure 3). Thus, lineage switching that occurs in this situation is formal rather than reflecting real biological changes.

Overall, the vast majority of recurrences occurred very early, which is in agreement with previously published data and with the statement that the main mechanism of ETP-ALL therapy escape is the change in lineage differentiation. At the same time, the boy with relapse 5 years after beginning frontline treatment also displayed a “lineage switch”, confirming that this is the main mechanism for chemoresistance in ETP-ALL, irrespective of the time to relapse/progression. We have to admit that the basic immunophenotypic definition of ETP-ALL is not extremely precise and includes patients with more or less typical non-ETP T-ALL in this group [22]. Hence, additional attempts to modify this immunophenotypic definition [23,24,25] or improve it with additional genetic studies have been published [3]. Nevertheless, nearly all recurrences occur in “true” early leukemia cases sharing T-lineage and myeloid features and result in final changes in diagnosis at relapse/progression.

While lacking a particular unifying genetic event, ETP-ALL is enriched in mutations and translocations involving genes related to transcriptional regulation, as noted in approximately 89% of cases, including *BCL11B*, *ETV6*, *RUNX1*, and *KMT2A,* etc. [26]. In our primary cohort, *BCL11B* rearrangement was the most frequent recurrence event, occurring in 10 of 60 ETP-ALL cases. We also observed translocations involving *ETV6*, *RUNX1*, *KMT2A*, and *MLLT10*, which are miscellaneous transcription factors and epigenetic regulators involved in definitive hematopoiesis [27,28,29]. Thus, their disruption intrinsically blocks the differentiation of early progenitor cells and promotes the leukemic phenotype throughout multiple AL subtypes [30]. These results are further supported by *TCR*/*BCR* clonal rearrangement repertoire analysis. Here, we report two patterns, each accounting for approximately half of the primary cases: polyclonal repertoire and TRD clonal rearrangements. However, relapsed ETPs were not enriched in any particular molecular pattern. The relapsed cohort included patients carrying both recurrent and nonrecurrent chromosomal aberrations as well as polyclonal and oligoclonal *TCR*/*BCR* clonal repertoires. Of note, ETP-ALL with polyclonal leukemic cells at diagnosis would likely result in AML or MPAL at relapse. At relapse, leukemic cells remain genetically unchanged. Nevertheless, the protein set responsible for the immunophenotype is altered. This further suggests that the immunophenotypic lineage switch in ETP-ALL relapse does not have a significant genetic basis and conforms to classification criteria rather than reflects biological changes.

## 4. Patients and Methods

### 4.1. Patients

Between 2017 and 2022, 7518 patients were diagnosed with acute leukemia (AL) at the Laboratories of the Dmitry Rogachev National Medical Research Center of Pediatric Hematology, Oncology and Immunology. Leukemic cells were characterized at initial diagnosis and relapse using a range of techniques, including immunophenotyping, fluorescent in situ hybridization (FISH), conventional karyotyping, next-generation sequencing (NGS) of the clonal T-cell receptor (TCR) repertoire, reverse transcription PCR (RT−PCR), and RNA sequencing (RNA-seq). Immunophenotyping was performed according to established guidelines [13]. ALL subtypes were defined per EGIL [31] with later modifications [10,13]. ETP-ALL was diagnosed with respect to the abovementioned criteria and reported in addition to TI (no specific T-lineage markers positive except CD7 and iCD3) or TII (CD2 or CD5 positive but without more mature T-lineage antigens) EGIL subtype [10]. Morphological and genetic studies were conducted according to national standards [32] using conventional techniques and as recommended by the manufacturers of kits and reagents.

### 4.2. Conventional Karyotyping and FISH 

Karyotyping was performed using G-banding. Metaphase analysis was performed using IKAROS software v.6.1.2 (Metasystems GMBH, Altlussheim, Germany). The probes used for the FISH study are listed in Appendix A.

### 4.3. RNA-seq and RT−PCR 

RNA-seq (Nextera UltraII Directional RNA kit, NEB, Ipswich, MA, USA) was performed using the Illumina NextSeq (Illumina, San Diego, CA, USA) platform, and fusion transcripts were analyzed using STAR aligner (ver. 2.10.7b) [33] and Arriba (ver. 2.4.0) [34]. RT−PCR was performed for quantitative assessment of *TLX3* expression and for qualitative assessment of *ETV6::MNX1* and *KMT2A::MLLT3* chimeric transcripts. The primers used are listed in Appendix A.

### 4.4. Assessment of the TCR Clonal Repertoire by NGS 

*TCR* rearrangements were evaluated by multiplex PCR for rearrangements of the *TCR* and *BCR* loci (*TRG*, *TRD*, *TRB*, *TRA IGH*, *IGK*, and *IGL*, MiLaboratory LLC., Sunnyvale, CA, USA, https://milaboratories.com/kits, accessed on 10 February 2022). Next, high-throughput NGS of the obtained amplicons was performed using an Illumina MiSeq instrument (Illumina, San Diego, CA, USA). The MiXCR pipeline [35] was used for bioinformatics analysis of the data. A frequency of 5% was applied as the cutoff for identifying leukemic clone-specific rearrangements. A repertoire with no clones exceeding 5% was considered polyclonal.

## 5. Conclusions

Thus, children with ETP-ALL typically experience relapse/progression with a change in lineage definition but retain cytogenetic and molecular features. Relapse of ETP-ALL requires thorough clinical and laboratory diagnostics with obligatory confirmation by molecular techniques [36]. Treatment decisions should rely mainly on the comprehensive evaluation of the initial examination data while also considering both immunophenotypic and molecular/genetic characteristics of leukemic cells and should probably be augmented with minimal residual disease data [9].

## Figures and Tables

**Figure 1 ijms-25-05610-f001:**
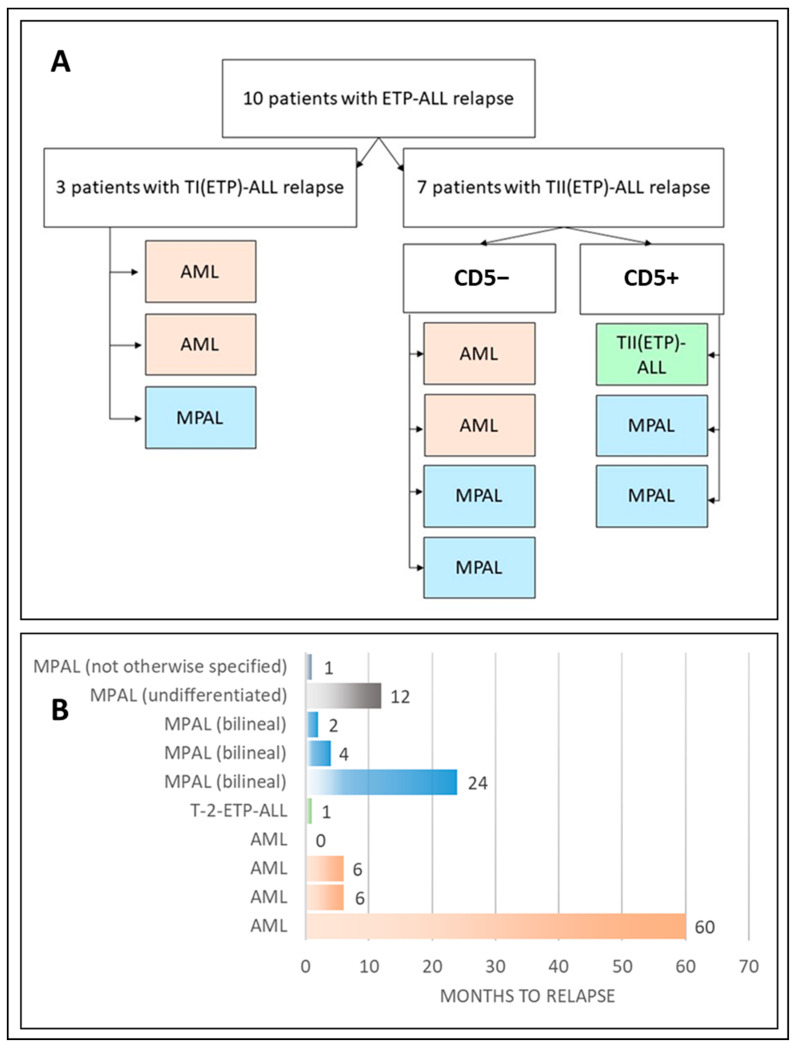
Relapse/progression pattern of pediatric ETP-ALL. Panel (**A**) shows the distribution of immunophenotypic variants of ETP-ALL relapse with respect to the initial diagnosis. White background, immunophenotypes at initial study; colored background, immunophenotypes at relapse. Panel (**B**) displays the time from diagnosis to relapse/progression, indicating acute leukemia type at relapse.

**Figure 2 ijms-25-05610-f002:**
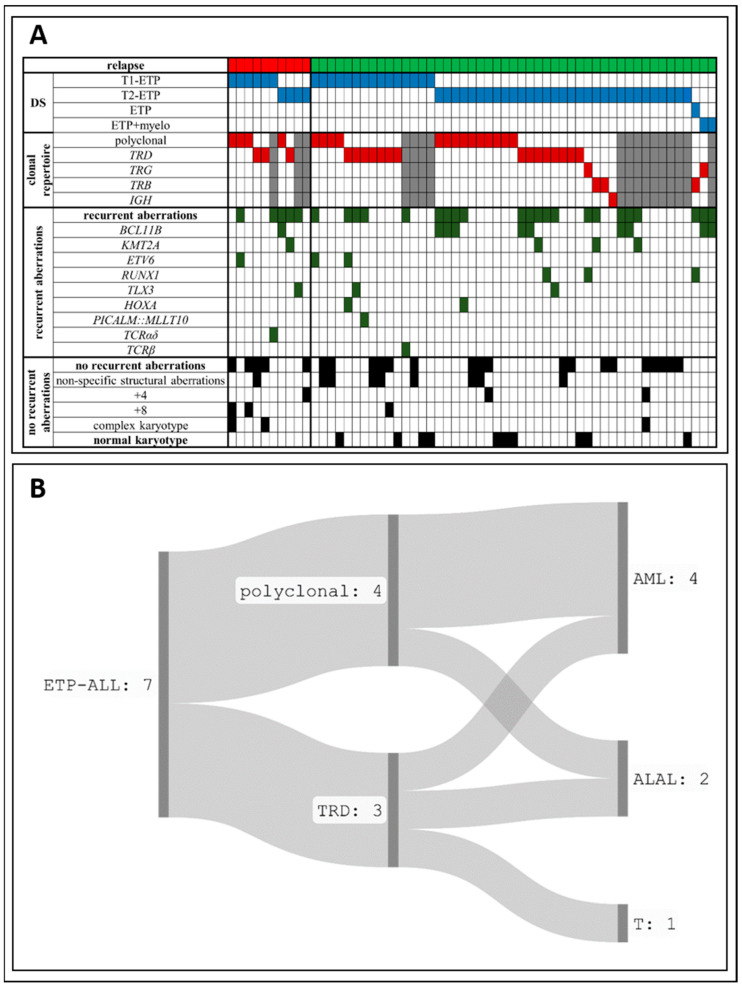
Genetic characterization of the described cases. Panel (**A**) shows genetic lesions and clonal *TCR*/*BCR* rearrangements at the initial examination in ETP-ALL with subsequent relapse (n = 10) and with no evidence of relapse at the time of the study (n = 50). Colored squares represent positive results, white–negative, gray–assessment not performed. Panel (**B**) displays the *TCR*/*BCR* repertoire of leukemic populations at initial examination of ETP-ALL and the type of acute leukemia at relapse.

**Figure 3 ijms-25-05610-f003:**
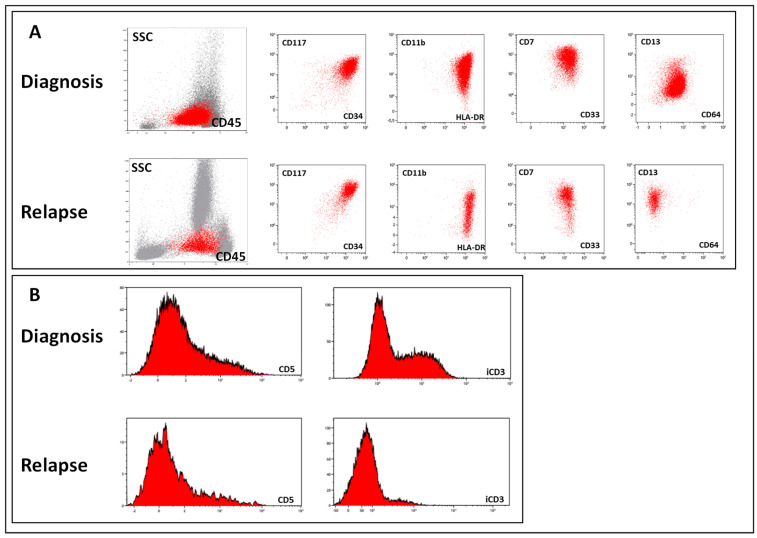
Example of a formal lineage switch from ETP-ALL (TII by EGIL) to AML at relapse. Panel (**A**) shows retaining key characteristic features of early cells with myeloid potential to differentiation. Panel (**B**) depicts a nearly complete loss of expression of diagnostic T-lineage antigens, resulting in a change in diagnosis from ETP-ALL to AML.

**Table 1 ijms-25-05610-t001:** Details of immunophenotypes at initial diagnosis and relapse of ETP-ALL.

Patient	Sex	Age y. m	Immunophenotype at Diagnosis	Time to Relapse, y, m, d	Immunophenotype at Relapse	Change in AL Type
1	m	16.1	CD5, CD7, CD11c, CD11b, CD13, CD33, CD34, CD38, CD45, iCD3, iCD79a	0, 1, 4	CD5, CD7, CD13, CD33, CD34, CD38, CD45, iCD3	TII(ETP)-ALL to TII(ETP)-ALL
2	m	4.8	CD5, CD7, CD33, CD34, CD45, CD117, HLA-DR, iCD3, iCD79a	1, 26	CD7, CD11a, CD33, CD34, CD38, CD45, CD99, CD117, CD123, HLA-DR	TII(ETP)-ALL to undifferentiated AL
3	m	6.1	CD2, CD7, CD8, CD11a, CD11c, CD11b, CD33, CD45, CD56, CD117, HLA-DR, iCD3, iCD79a	0, 4, 5	Population 1: 5% CD45^dim^/SSC^dim^: CD4, CD7, CD11a, CD11b, CD13, CD33, CD34, CD45, CD56, CD117, CD123, HLA-DR	TII(ETP)-ALL to bilineal MPAL
					Population 2: 10% CD45^high^/SSC^dim^: CD2, CD7, CD8, CD11a, CD11b, CD45, CD56, HLA-DR, iCD3	
4	f	0.6	CD7, CD11b, CD11c, CD13, CD33, CD34, CD45, CD56, CD117, HLA-DR, iCD3	0, 6, 27	CD7, CD11a, CD11b, CD13, CD15, CD33, CD34, CD38, CD45, CD56, CD99, CD117, CD123, CD371, HLA-DR	TI(ETP)-ALL to AML
5	f	6.2	CD2, CD3, CD7, CD33, CD34, CD45, CD117, iCD3	0, 0, 25	CD2, CD3, CD7, CD33, CD34, CD45, CD117	TII(ETP)-ALL to AML
6	m	8.6	CD7, CD11a, CD11c, CD33, CD34, CD45, CD99, CD117, HLA-DR, iCD3, iCD79a	0, 6, 0	CD7, CD11a, CD11b, CD11c, CD13, CD15, CD33, CD34, CD38, CD45, CD99, CD123, HLA-DR, CD117, CD123, CD371	TI(ETP)-ALL to AML
7	m	5.6	CD2, CD7, CD11b, CD13, CD33, CD34, CD45, CD56, CD117, HLA-DR, iCD3	5, 0, 0	CD11b, CD33, CD34, CD38, CD45, CD56, CD99, CD117, CD123	TII(ETP)-ALL to AML
8	m	15.9	CD3, CD5, CD7, CD13, CD15, CD34, CD38, CD45, CD99, CD117, iCD3	0, 2, 27	Population 1: 11% CD45^high^/SSC^high^: CD4, CD11a, CD11c, CD11b, CD33, CD34, CD38, CD45, CD56, CD64, CD117, CD133,	TII(ETP)-ALL to bilineal MPAL
					Population 2: 13% CD45^dim^/SSC^dim^: CD7, CD38, CD45, CD117, iCD3	
9	m	13.0	CD7, CD22, CD34, CD45, CD117, HLA-DR, iCD3	0, 1, 24	CD4, CD7, CD11c, CD34, CD38, CD45, CD64, CD117, HLA-DR	TI(ETP)-ALL to unclassifiable AL
10	m	14.8	CD2, CD7, CD11a, CD13, CD34, CD45, CD117, HLA-DR, iCD3	2, 3	Population 1: 83%, CD2, CD7, CD11a, CD13, CD15, CD33, CD34, CD38, CD45, CD99, CD117, CD371, HLA-DR, iCD3	TII(ETP)-ALL to bilineal MPAL
					Population 2: 5%, CD2, CD13, CD15, CD33, CD117, CD371, MPO	

**Table 2 ijms-25-05610-t002:** Molecular and genetic features of leukemic cells at initial diagnosis and relapse of ETP-ALL.

Patient	Sex	Age, y.m	Timepoint	DS	Clones	Cytogenetics
1	m	16.1	diagnosis	TII(ETP)-ALL	*TRD D2_D3*	del17p within complex karyotype
			relapse	TII(ETP)-ALL	*TRD D2_D3*	del17p within complex karyotype
2	m	4.8	diagnosis	TII(ETP)-ALL	*TRD V1_D3_J1*	*KMT2A::MLLT3*
			relapse	undifferentiated AL	*TRD V1_D3_J1*	*KMT2A::MLLT3* within complex karyotype
3	m	6.1	diagnosis	TII(ETP)-ALL	polyclonal	trisomy 8
			relapse	MPAL	polyclonal	trisomy 8
4	f	0.6	diagnosis	TI(ETP)-ALL	polyclonal	*MNX1::ETV6*
			relapse	AML	polyclonal	*MNX1::ETV6*
5	f	6.2	diagnosis	TII(ETP)-ALL	polyclonal	*BCL11B*
			relapse	AML	polyclonal	*BCL11B*
6	m	8.6	diagnosis	TI(ETP)-ALL	polyclonal	complex
			relapse	AML	polyclonal	complex
7	m	5.6	diagnosis	TII(ETP)-ALL	*TRD D2_D3*	trisomy 10
			relapse	AML	*TRD D2_D3*	trisomy 10
8	m	15.9	diagnosis	TII(ETP)-ALL	no data	*TLX3*
			relapse	MPAL	no data	*TLX3*
9	m	13.0	diagnosis	TI(ETP)-ALL	no data	complex
			relapse	unclassifiable AL	no data	complex, del17p
10	m	14.8	diagnosis	TII(ETP)-ALL	no data	trisomy 4
			relapse	MPAL	no data	no data

## Data Availability

The datasets generated during and/or analyzed during the current study are available from the corresponding author upon reasonable request. RNAseq data is available in the GEO database (https://www.ncbi.nlm.nih.gov/geo/ accessed on 1 May 2024) under accession number GSE266550.

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
