# Peer review of "Immunophenotypic but Not Genetic Changes Reclassify the Majority of Relapsed/Refractory Pediatric Cases of Early T-Cell Precursor Acute Lymphoblastic Leukemia"

_ijms, 2024, doi:10.3390/ijms25115610_

Round 1
Reviewer 1 Report
Comments and Suggestions for Authors
The article from Demina et al. is a well-written study that describes the authors' findings after gathering data from a large number of T-ALL patients. The article is clear and concise and directly addresses important points related to the ETP-ALL, well known for its difficulty in properly diagnosing it and for its worse prognosis. The study focuses on describing the ETP relapses, discussing and noting features that lead to the reclassification of the diseases after the relapse, and considering the features lost and/or acquired. This brings valuable data and information to the table and a study like this will be pivotal for defining the landscape of ETP diagnostics and treatment. I commend the authors for the well-conducted study and the writing of this article.
As such, I did not find any major concern with the article and I find it highly informative and potentially ready for publication. I have some minor points that I can share with the authors, which are easily addressable.
1) The abbreviation of Pat in the tables is not a standard one, I suggest using the full "Patient" word.
2) At line 82 the authors consider TI and TII ALL, but this was not introduced and described previously. I suggest doing so, as some readers could be less familiar with the field and this could improve clarity for them.
3) The authors mention RNA-seq, but classical RNA-seq analysis with heatmaps, DEG, pathway analysis, etc. was not shown. Data is available upon request. Maybe this will be analyzed in a different article? Generally, I suggest to make genomic data available in GEO
As previously stated, this article is in very good shape and these minor points do not impact its overall quality.
Author Response
We thank the Reviewer for so favorable review of our manuscript..
Regarding the comments mentioned:
- we have corrected both tables
- we have added these definitions both to the methods section and to mentioned lines in the results.
- We thank the reviewer for this comment. Unfortunately, only a subset of samples were subjected to RNAseq due to limited material availability. Thus, we were unable to perform classical differential gene expression analysis or pathway enrichment analysis and only searched for fusion transcripts. However, we have uploaded our RNAseq data to GEO, they are available under accession number GSE266550. This statement is added also to the Data sharing statement
Reviewer 2 Report
Comments and Suggestions for Authors
Dear editor,
Thank you for the opportunity to revise the paper ''Immunophenotypic but not genetic changes reclassify the majority of relapsed/refractory pediatric cases of early T-cell precursor acute lymphoblastic leukemia''. ETP-ALL is rare enthity and the data regarding this disease are scarce. Therefore the topic is actual and interesting for the readers. The study is well structured and supported with the references from the literature. The methods are well described. And the results are clearly depicted.
Author Response
We thank the Reviewer for so favorable comments on our work